# Antibiotic dispensing practices and antimicrobial stewardship gaps in community pharmacies in Kakamega County, Kenya

Eleanor Turnbull-Jones[1,2*], Susannah Langtree[1], Nicholas Mogoi[3], Anthony Sifuna[3], Lindsay Gadaffi[4], Tony Jewell[1]

**1** Cambridge Global Health Partnerships, Cambridge, United Kingdom, **2** Cambridge University Hospitals NHS Foundation Trust, Cambridge, United Kingdom, **3** Masinde Muliro University of Science and Technology, Kakamega, Kenya, **4** Ministry of Health Services, Kakamega, Kenya

\* et605@cam.ac.uk

## Abstract

### Background

Antimicrobial resistance (AMR) is a major global health threat, with sub-Saharan Africa bearing a disproportionate burden. Community-level antibiotic dispensing practices remain poorly described in Kenya outside Nairobi.

### Methods

A total of 504 antibiotic dispensing events were recorded across 22 community pharmacies in Kakamega County, western Kenya, between 3rd and 22nd August 2025. Data collected included dispensing source (over-the-counter [OTC] versus prescription), clinical indication, antibiotics dispensed, course completion, and self-reported repeat antibiotic use within the preceding month. Descriptive analyses were performed, and $\chi^2$ tests were used to examine associations between dispensing source and selected non-antibiotic dispensing characteristics.

### Results

Of the 504 dispensing events, 224 (44.4%) involved OTC dispensing and 278 (55.2%) were prescription-based. The most frequent indications for antibiotic dispensing were upper respiratory tract infections (URTI; n = 156, 31.0%), lower respiratory tract infections (LRTI; n = 95, 18.8%), gastrointestinal infections (n = 65, 12.9%), and skin or soft-tissue infections (n = 55, 10.9%). Across all events, amoxicillin, azithromycin, and metronidazole were the most frequently dispensed antibiotics, with cephalosporins and other broad-spectrum agents used across several indications. Partial antibiotic courses were supplied in 33 (6.5%) dispensing events, most commonly due to financial constraints (15/33, 45.5%). Self-reported antibiotic use within the preceding month occurred in 156 (31.0%) cases.

**Data availability statement:** The anonymised dataset and supporting documentation are available in Figshare, 10.6084/m9.figshare.31078666.

**Funding:** This work was supported by the Commonwealth Partnerships for Antimicrobial Stewardship (CwPAMS), funded by the UK Department of Health and Social Care (DHSC) through the Fleming Fund [project reference: CwPAMS 2.5 A.01], and by Cambridge Global Health Partnerships (CGHP). The funders had no role in study design, data collection and analysis, decision to publish, or preparation of the manuscript.

**Competing interests:** The authors have declared that no competing interests exist.

## Conclusions

OTC antibiotic access remains widespread in Kakamega County, with substantial use of broad-spectrum agents across multiple clinical indications. Financial barriers contribute to incomplete antibiotic courses. These findings highlight the importance of incorporating community pharmacy dispensing data into county-level antimicrobial stewardship programmes and informing national strategies to optimise antibiotic use.

## Background

Antimicrobial resistance (AMR) remains one of the leading global public health threats, directly responsible for an estimated 1.27 million deaths and associated with 4.95 million deaths globally in 2019, based on the most comprehensive analysis to date. The burden is highest in sub-Saharan Africa [1] The inappropriate use of antibiotics—particularly self-medication, incomplete courses, and dispensing without prescription—accelerates the emergence of resistant pathogens [2].

In Kenya, community pharmacies are a major source of antibiotics for both prescribed and non-prescribed use [3] These pharmacies represent a key interface between health systems and communities, often serving as the first point of care for common infections, particularly where access to formal healthcare is limited. While the Pharmacy and Poisons Act restricts antibiotic sales without prescription, enforcement remains inconsistent, particularly in peri-urban and rural settings. Evidence from Nairobi County indicates that 20–30% of antibiotic transactions occur without valid prescriptions [4], yet data from other counties are scarce. Limited surveillance outside major urban centres hampers local implementation of antimicrobial stewardship (AMS) interventions.

To strengthen containment of AMR, Kenya launched the National Action Plan on Antimicrobial Resistance (2023–2027), emphasising surveillance, rational antibiotic use, and integration of community-level data [5]. The Commonwealth Partnerships for Antimicrobial Stewardship (CwPAMS) programme also supports pharmacy-led AMS initiatives across Kenyan counties. However, little is known about how antibiotics are dispensed at the community level outside Nairobi, and whether dispensing patterns reflect the intended AMS goals. Understanding real-world dispensing practices is a necessary first step for designing contextually appropriate antimicrobial stewardship interventions in community pharmacy settings.

This study therefore describes antibiotic dispensing practices in Kakamega County, western Kenya—an area representing rural and peri-urban pharmacy settings. We characterise the indications for antibiotic dispensing, patterns of over-the-counter (OTC) versus prescription dispensing, and factors associated with partial and repeat antibiotic courses, to inform local AMS policy and national surveillance priorities.

## Methods

### Study design and setting

This was a cross-sectional study conducted across 22 community pharmacies in Kakamega County, western Kenya, between 3rd and 22nd August 2025. The county

represents a mix of urban, peri-urban, and rural settings. Participating outlets were licensed under the Pharmacy and Poisons Board (PPB) and included both independent and chain pharmacies. All eligible dispensing events during the study period were included consecutively to minimise selection bias and capture real-world dispensing practices. A formal sample size calculation was not performed as the study aimed to include all available dispensing events within the defined time frame.

## Inclusion and exclusion criteria

All consecutive dispensing events involving at least one systemic human antibiotic, irrespective of formulation or duration, during the study period were included. Dispensing of topical, veterinary, or non-systemic antimicrobial agents was excluded. Each entry represented a single antibiotic dispensing event.

## Data collection

Pharmacy staff recorded antibiotic dispensing events using a structured data collection tool. Variables captured included dispensing source (over-the-counter [OTC] or prescription), clinical indication (as reported by the patient or assessed by the dispenser), antibiotic(s) dispensed, provider of indication (doctor, pharmacist or pharmacy technician, patient, or other), whether a full antibiotic course was supplied and reasons for partial courses, and self-reported antibiotic use within the previous month. Where no prescription was presented, the clinical indication reflected either patient self-report or assessment by pharmacy staff. No personally identifiable patient information was collected.

## Data management and cleaning

Data were entered into Microsoft Excel and exported to Python (version 3.10) for cleaning and analysis. Free-text variables were standardised to ensure consistent spelling and grouping. Forty-seven age entries that were erroneously parsed as dates during data export were manually recoded to the appropriate 5–12 years age category prior to analysis. Clinical indications were grouped into eight categories: upper respiratory tract infection (URTI), lower respiratory tract infection (LRTI), skin or soft-tissue infection (SSTI), gastrointestinal infection (GI), urinary tract infection (UTI), dental infection, sexually transmitted infection (STI), and other or non-specific indications. Antibiotic names were standardised and grouped using individual drug names for commonly dispensed agents and clinically relevant antibiotic classes, informed by the World Health Organization's AWaRe framework. Where more than one antibiotic was dispensed during a single encounter, each antibiotic was included separately in descriptive analyses of antibiotic use.

## Statistical analysis

Descriptive statistics were used to summarise frequencies and percentages. Associations between categorical variables—including dispensing source, clinical indication, provider of indication, age group, partial course dispensing, and repeat antibiotic use—were evaluated using Pearson's chi-square ($\chi^2$) test. Statistical significance was defined as $p < 0.05$. Analyses were performed using Python version 3.10 with the SciPy library (version 1.11.3).

This study adheres to the STROBE guidelines for cross-sectional studies.

This study was approved by the Masinde Muliro University of Science and Technology Institutional Scientific and Ethics Review Committee (REF: MMU/COR: 40312 Vol 6(01)) and the National Commission for Science, Technology and Innovation (NACOSTI) (License No: NACOSTI/P/25/4176035). All participating pharmacists and technicians provided verbal informed consent to join the study. No personally identifiable data were collected, and the ethics committees granted a waiver of individual patient consent due to the use of fully anonymised routine dispensing data. Additional information regarding the ethical, cultural, and scientific considerations specific to inclusivity in global research is included in the Supporting Information (S2 File).

## Results

### Dispensing source

A total of 504 antibiotic dispensing events were recorded across 22 community pharmacies in Kakamega County. Of these, 224 (44.4%) involved over-the-counter (OTC) dispensing, 278 (55.2%) were prescription-based, and 2 (0.4%) dispensing events involved cases where a prescription was presented but additional antibiotics were requested and supplied without prescription.

### Clinical indications for antibiotic dispensing

Using the predefined indication groupings, the most frequent indications for antibiotic dispensing were upper respiratory tract infections (URTI; n = 156, 31.0%), followed by lower respiratory tract infections (LRTI; n = 95, 18.8%), gastrointestinal infections (GI; n = 65, 12.9%), and skin or soft-tissue infections (SSTI; n = 55, 10.9%). Other indications included sexually transmitted infections (STI; n = 33, 6.5%), dental infections (n = 27, 5.4%), urinary tract infections (UTI; n = 20, 4.0%), sepsis (n = 11, 2.2%), and other or non-specific indications (n = 80, 16.5%) (Table 1). The distribution of clinical indications did not differ significantly by dispensing source ($\chi^2$ = 18.47, df = 16, p = 0.297).

### Provider of clinical indication

Doctors were the most frequent source of antibiotic indications for dental infections, UTIs, and sepsis, while pharmacists or pharmacy technicians most commonly provided indications for LRTIs and SSTIs. Patients most frequently self-identified the need for antibiotics for URTIs and other non-specific indications. The association between indication category and source of indication was statistically significant ($\chi^2$ = 54.84, df = 32, p = 0.007). The distribution of the source of clinical indication across infection categories is illustrated in Fig 1.

### Antibiotics dispensed

Across all dispensing events, a total of 558 antibiotics were dispensed, reflecting that some encounters involved more than one antibiotic. Of the 504 dispensing events, 52 (10.3%) involved more than one antibiotic, with the majority involving two antibiotics and very few cases involving three or more. The most frequently dispensed antibiotics overall were amoxicillin (n = 92), cephalosporins (n = 85), azithromycin (n = 74), fluoroquinolones (n = 53), metronidazole (n = 52), and

**Table 1. Clinical indications for antibiotic dispensing by age group (n = 485).**

| Indication | <5 years | 5–12 years | 13–19 years | 20–64 years | ≥65 years | Total n |
|---|---|---|---|---|---|---|
| Upper respiratory tract infection (URTI) | 19 | 18 | 27 | 84 | 8 | 156 |
| Lower respiratory tract infection (LRTI) | 10 | 7 | 10 | 60 | 8 | 95 |
| Gastrointestinal infection (GI) | 6 | 5 | 7 | 43 | 4 | 65 |
| Skin and soft-tissue infection (SSTI) | 8 | 4 | 5 | 34 | 4 | 55 |
| Sexually transmitted infection (STI) | 1 | 0 | 6 | 24 | 2 | 33 |
| Dental infection | 1 | 3 | 2 | 19 | 2 | 27 |
| Urinary tract infection (UTI) | 2 | 2 | 3 | 11 | 2 | 20 |
| Sepsis | 1 | 0 | 1 | 7 | 2 | 11 |
| Other/ non-specific indications | 11 | 8 | 27 | 29 | 5 | 80 |
| Total n (%) | 59 (12.2) | 47 (9.7) | 88 (18.1) | 251 (51.8) | 40 (8.2) | 485 (100) |

Notes:

Age data were available for 485 of 504 dispensing events; events with missing age data were excluded from age-stratified analyses. Forty-seven age entries originally parsed as dates were manually recoded to the correct 5–12 years category prior to analysis. Values are presented as frequencies (n) for each indication. Percentages are shown only for age group totals and are calculated based on the number of dispensing events with available age data (n = 485).

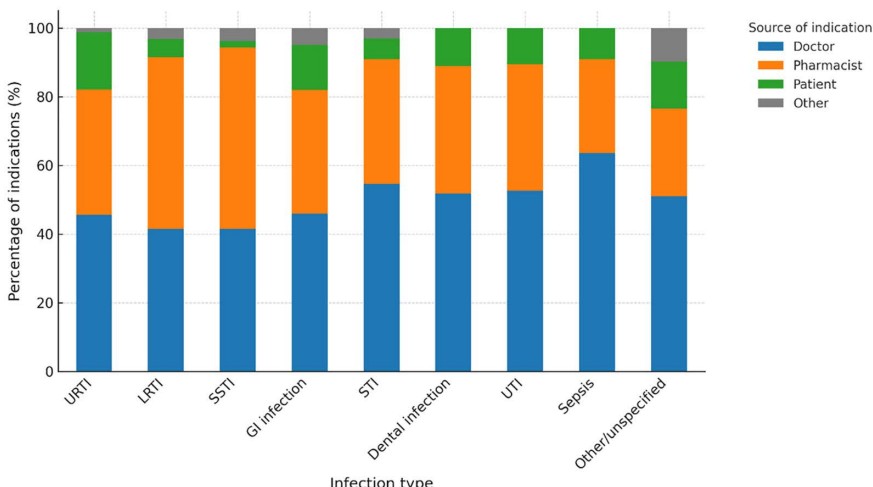

**Fig 1. Source of clinical indication for antibiotic dispensing by infection type.** Stacked bar chart showing the proportion of antibiotic dispensing events by source of clinical indication (doctor, pharmacist or pharmacy technician, patient, or other) across major infection categories in community pharmacies in Kakamega County, western Kenya. Percentages are calculated within each infection category.

co-amoxiclav (n = 46). According to the WHO AWaRe classification, the majority of antibiotics dispensed belonged to the Access and Watch categories, with frequent use of Watch antibiotics such as macrolides and fluoroquinolones.

## Antibiotic selection by clinical indication

Antibiotic choice varied by clinical indication (Table 2). Macrolides were the most frequently dispensed antibiotics for URTIs, closely followed by penicillins. For LRTIs, penicillins were the most commonly used antibiotics, followed by cephalosporins, with smaller contributions from macrolides and other antibiotic classes. Nitroimidazoles predominated in gastro-intestinal infections, with substantial additional use of fluoroquinolones. Antibiotic use for SSTIs was heterogeneous, with a wide range of agents included within the "other" category, alongside use of nitroimidazoles and penicillins. Dental infections were most commonly treated with penicillins, often in combination with nitroimidazoles. Antibiotic selection for STIs and UTIs was heterogeneous, with contributions from cephalosporins, fluoroquinolones, tetracyclines, and other agents.

## Age distribution

Age data were available for 485 of 504 (96.2%) dispensing events. Among these, 59 (12.2%) involved children aged <5 years, 47 (9.7%) aged 5–12 years, 88 (18.1%) aged 13–19 years, 251 (51.8%) aged 20–64 years, and 40 (8.2%) aged ≥65 years. The distribution of clinical indications by age group is presented in Table 1.

## Partial antibiotic courses

Partial antibiotic courses were dispensed in 33 of 504 (6.5%) events. The most frequently reported reason was financial constraint (15/33, 45.5%), followed by patient preference for reduced quantities and stock shortages. There was no significant association between partial course dispensing and dispensing source ($\chi^2 = 1.94$, df = 2, p = 0.38).

## Repeat antibiotic use

Self-reported antibiotic use within the preceding month occurred in 156 of 504 (31.0%) dispensing events. Among these, 84 (53.8%) were repeats for the same condition, 49 (31.4%) for a different condition, and 23 (14.7%) were unspecified. Repeat use did not differ significantly by dispensing source ($\chi^2 = 0.83$, df = 2, p = 0.66).

**Table 2. Antibiotics dispensed by clinical indication.**

| Indication | Antibiotic class | n |
|---|---|---|
| URTI | Macrolides | 51 |
| | Penicillins | 47 |
| | Other | 42 |
| | Cephalosporins | 18 |
| | Nitroimidazoles | 3 |
| | Fluoroquinolones | 1 |
| LRTI | Penicillins | 50 |
| | Cephalosporins | 24 |
| | Other | 17 |
| | Macrolides | 11 |
| | Nitroimidazoles | 1 |
| | Fluoroquinolones | 2 |
| GI | Nitroimidazoles | 30 |
| | Fluoroquinolones | 24 |
| | Cephalosporins | 5 |
| | Macrolides | 5 |
| | Other | 5 |
| | Penicillins | 2 |
| SSTI | Other | 53 |
| | Nitroimidazoles | 11 |
| | Penicillins | 6 |
| | Macrolides | 2 |
| | Cephalosporins | 1 |
| UTI | Fluoroquinolones | 8 |
| | Other | 8 |
| | Penicillins | 3 |
| | Cephalosporins | 3 |
| | Macrolides | 2 |
| STI | Cephalosporins | 25 |
| | Other | 20 |
| | Nitroimidazoles | 10 |
| | Macrolides | 6 |
| Dental | Penicillins | 21 |
| | Other | 11 |
| | Nitroimidazoles | 7 |
| Sepsis | Other | 9 |
| | Cephalosporins | 3 |
| | Nitroimidazoles | 2 |
| | Macrolides | 1 |
| Other indications | Other | 31 |
| | Fluoroquinolones | 18 |
| | Nitroimidazoles | 10 |
| | Penicillins | 8 |
| | Cephalosporins | 8 |
| | Macrolides | 3 |

**Notes:**

Counts represent the total number of times each antibiotic class was dispensed within each clinical indication category. Dispensing events involving more than one antibiotic are counted separately for each antibiotic. Antibiotics were grouped into classes based on pharmacological classification informed by the WHO AWaRe framework.

## Discussion

This study provides one of the first detailed descriptions of antibiotic dispensing practices in community pharmacies outside Nairobi, Kenya. Across 22 outlets in Kakamega County, nearly half of all antibiotics were dispensed without a prescription, underscoring the continued accessibility of antibiotics through non-regulated channels. These findings are consistent with prior studies from sub-Saharan Africa reporting high levels of non-prescription antibiotic sales; however, the 44.4% OTC dispensing rate observed here exceeds estimates reported from urban Kenyan settings, suggesting more limited regulatory enforcement in peri-urban and rural areas [6–8].

The antibiotics dispensed were dominated by amoxicillin, azithromycin, and metronidazole, reflecting largely empiric antibiotic use in community settings where access to diagnostic testing is limited. Similar patterns have been reported across East Africa, where antibiotic choice is often driven by symptom-based assessment rather than microbiological confirmation [9]. According to the WHO AWaRe classification, a substantial proportion of antibiotics dispensed belonged to the Access and Watch categories. Of particular concern is the frequent use of macrolides and fluoroquinolones, which fall within the Watch category and are recommended for more restricted use due to their higher potential to drive antimicrobial resistance [5]. Detailed patterns of antibiotic class use across indications are presented in Table 2.

Respiratory tract infections, including both URTIs and LRTIs, were the most common indications for antibiotic dispensing, accounting for nearly half of all events, consistent with prior research highlighting respiratory conditions as key drivers of antibiotic overuse in outpatient settings [10]. Antibiotic selection for URTIs was dominated by macrolides and penicillins, while LRTIs were most commonly treated with penicillins, followed by cephalosporins. This pattern suggests a combination of guideline-consistent prescribing alongside empiric escalation to broader-spectrum agents in some cases.

In gastrointestinal infections, nitroimidazoles—particularly metronidazole—predominated, alongside substantial use of fluoroquinolones. While metronidazole use is consistent with treatment of certain parasitic and anaerobic infections, the concurrent use of fluoroquinolones may reflect empiric treatment approaches that do not always align with guideline-recommended therapy.

Antibiotic use for skin and soft-tissue infections was heterogeneous, reflecting variability in prescribing practices and potential uncertainty in diagnosis or management at the community level. Dental infections were most commonly treated with penicillins, often in combination with nitroimidazoles, consistent with common clinical practice in settings where mixed infections are suspected.

Pharmacists and pharmacy technicians played a central role in determining clinical indications for antibiotic use, particularly for LRTIs and SSTIs, while doctors most frequently provided indications for dental infections, UTIs, and sepsis. This pattern reflects the expanding clinical role of pharmacists in community settings, especially in contexts where access to formal medical care may be constrained. Similar findings have been reported in Ghana and Nigeria, where pharmacy staff frequently guide antibiotic selection in the absence of a prescription [11–13]. These results highlight an important opportunity to integrate community pharmacists more formally into county-level antimicrobial stewardship (AMS) programmes, including through targeted training, clinical decision support, and strengthened referral pathways.

Incomplete antibiotic courses were dispensed in 6.5% of events, most commonly due to financial constraints, underscoring persistent socioeconomic barriers to full-course adherence. Although the proportion of partial courses was modest, even short interruptions or under-dosing may contribute to treatment failure and selection for resistance. These findings mirror reports from other low- and middle-income countries, where cost remains a major determinant of antibiotic adherence [14] Community AMS strategies must therefore balance efforts to restrict inappropriate use with measures to ensure affordable access to complete antibiotic courses.

Repeat antibiotic use within the preceding month was reported in nearly one-third of dispensing events, often for the same condition. While repeat use was not significantly associated with dispensing source, such recurrent exposure increases the risk of antimicrobial resistance and may indicate unresolved infections, inadequate prior treatment, or

 

inappropriate initial antibiotic selection. Strengthening patient counselling, improving continuity of care, and exploring mechanisms for shared dispensing records across pharmacies could help reduce unnecessary repeat antibiotic use.

Overall, these findings reveal persistent gaps in antibiotic governance at the community level and underscore the central role of pharmacists in antibiotic selection and supply. Addressing inappropriate antibiotic use in settings such as Kakamega County will require strengthening regulatory enforcement alongside pragmatic, pharmacy-led stewardship interventions. Integrating community pharmacy dispensing data into Kenya's AMR surveillance systems and National Action Plan on AMR (2023–2027) is essential to support locally responsive and sustainable AMS strategies.

## Conclusion

Nearly half of antibiotics dispensed in community pharmacies in Kakamega County were supplied without a prescription, most commonly for respiratory tract infections. Amoxicillin, azithromycin, and metronidazole dominated antibiotic use, with substantial reliance on broad-spectrum agents across multiple indications. Incomplete antibiotic courses were frequently linked to financial constraints, and repeat antibiotic use was common. These findings highlight critical gaps in community antibiotic governance, including widespread non-prescription dispensing, reliance on broad-spectrum antibiotics, and barriers to completing treatment courses. The central role of pharmacists in determining antibiotic use further underscores the need for targeted antimicrobial stewardship interventions at the community level. Incorporating community pharmacy data into county-level surveillance and national AMR frameworks will be essential to optimise antibiotic use and mitigate the emergence of antimicrobial resistance.

## Supporting information

**S1 Table. Reasons for partial antibiotic course dispensing.** Summary of reported reasons for dispensing incomplete antibiotic courses across community pharmacies in Kakamega County, Kenya.
(DOCX)

**S2 File. Inclusivity in Global Research Questionnaire.** Completed PLOS ONE questionnaire on inclusivity and best practices in global research.
(DOCX)

## Author contributions

**Conceptualization:** Eleanor Turnbull-Jones, Susannah Langtree, Tony Jewell.

**Data curation:** Eleanor Turnbull-Jones.

**Formal analysis:** Eleanor Turnbull-Jones.

**Funding acquisition:** Susannah Langtree, Tony Jewell.

**Investigation:** Nicholas Mogoi, Anthony Sifuna, Lindsay Gadaffi.

**Methodology:** Eleanor Turnbull-Jones, Anthony Sifuna, Lindsay Gadaffi.

**Project administration:** Eleanor Turnbull-Jones, Tony Jewell.

**Supervision:** Tony Jewell.

**Visualization:** Susannah Langtree.

**Writing – original draft:** Eleanor Turnbull-Jones.

**Writing – review & editing:** Eleanor Turnbull-Jones, Susannah Langtree, Nicholas Mogoi, Anthony Sifuna, Lindsay Gadaffi, Tony Jewell.

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
