## [Decision Letter · Decision Letter 0]

31 Mar 2026

PONE-D-26-02714Antibiotic dispensing practices and antimicrobial stewardship gaps in community pharmacies in Kakamega County, KenyaPLOS One

Dear Dr. Turnbull-Jones,

Thank you for submitting your manuscript to PLOS ONE. After careful consideration, we feel that it has merit but does not fully meet PLOS ONE’s publication criteria as it currently stands. Therefore, we invite you to submit a revised version of the manuscript that addresses the points raised during the review process.

Kindly address the comments from the two reviewers below

Please submit your revised manuscript by May 15 2026 11:59PM. If you will need more time than this to complete your revisions, please reply to this message or contact the journal office at plosone@plos.org. Please include the following items when submitting your revised manuscript:

We look forward to receiving your revised manuscript.

Kind regards,

Eric Ogola, MPH

Academic Editor

PLOS One

Journal Requirements:

Reviewers' comments:

Reviewer's Responses to Questions

**Comments to the Author**

1. Is the manuscript technically sound, and do the data support the conclusions?

Reviewer #1: Yes

Reviewer #2: Yes

2. Has the statistical analysis been performed appropriately and rigorously? 

Reviewer #1: Yes

Reviewer #2: Yes

3. Have the authors made all data underlying the findings in their manuscript fully available?

Reviewer #1: Yes

Reviewer #2: Yes

4. Is the manuscript presented in an intelligible fashion and written in standard English?

Reviewer #1: Yes

Reviewer #2: Yes

5. Review Comments to the Author

Reviewer #1: 1. Antimicrobial resistance (AMR) remains one of the leading global public health threats,

70 contributing to an estimated 4.95 million deaths in 2019, with the highest burden in sub-

71 Saharan Africa. Please re check for latest data if available as per WHO , or any other valid agencies

2. major thing which is highlighted in conclusion is highlight critical gaps in community antibiotic governance. kindly highlight other important aspects of this study.

Reviewer #2: 1. The research and manuscript is well-structured. The manuscript can be accepted after minor revision since research like this may help to combat the menace of AMR in regions like Kenya.

2. The researchers mentioned in the methodology that they have followed STROBE guidelines for cross-sectional studies. But they have also described the study design as "prospective descriptive cross-sectional ".The design can be "cross-sectional study" since description of the methodology reveals that it is not prospective and since they have employed statistical tests also in interpreting data , the study is should be categorised as "analytical' rather than purely "descriptive".

3. For sampling, they have used consecutive prescription with antibiotic. But they have not provided a relevant justification for the sample "504" and the reason for choosing a particular sampling method.

4. Lines 154 - 155, they mentioned "both". How the dispensing source can be both OTC and prescription-based

5. Lines 178 - 179, they mentioned that few prescriptions had more than one antibiotic. But they did not mention how many prescriptions had two antibiotics, three antibiotics , etc.

6. In methodology and discussion they have included AwARe classification and it is missing in the results. To strengthen analysis, the authors should include the classification.

6. PLOS authors have the option to publish the peer review history of their article (what does this mean?). If published, this will include your full peer review and any attached files.

Reviewer #1: **Yes:** Dr. Saurabh Singh , Ex Professor, School of Pharmaceutical Sciences ,LPU

Reviewer #2: No

---

## [Author Response · Author response to Decision Letter 1]

21 Apr 2026

Dear Editors,

Thank you for the opportunity to revise and resubmit our manuscript entitled “Antibiotic dispensing practices and antimicrobial stewardship gaps in community pharmacies in Kakamega County, Kenya.” We are grateful to the editor and reviewers for their constructive and insightful comments, which have helped us to substantially strengthen the manuscript.

We have carefully addressed all editorial and reviewer comments and provide a detailed, point-by-point response below. All changes have been incorporated into the revised manuscript.

Editorial Comments

1. PLOS ONE formatting requirements

We have revised the manuscript to fully comply with PLOS ONE formatting guidelines, including structure, section headings, and file naming conventions. The manuscript has been aligned with the PLOS ONE formatting templates provided.

2. Inclusivity in global research questionnaire

A completed copy of the PLOS ONE inclusivity in global research questionnaire has been included as Supporting Information (S2 File), and this is referenced in the Methods section of the manuscript.

3. Ethics statement placement

The ethics statement has been reviewed and is now included exclusively within the Methods section. It has been removed from any other sections to comply with journal requirements.

4. Funding information consistency

The Funding statement has been reviewed and harmonised across all submission fields. All funding sources are now consistently reported in both the manuscript and submission system. The funding statement now reads, 'This work was supported by the Commonwealth Partnerships for Antimicrobial Stewardship (CwPAMS), funded by the UK Department of Health and Social Care (DHSC) through the Fleming Fund [project reference: CwPAMS 2.5 A.01], and by Cambridge Global Health Partnerships (CGHP). The funders had no role in study design, data collection and analysis, decision to publish, or preparation of the manuscript'.

5. Supporting Information captions

Captions for all Supporting Information files have been added at the end of the manuscript, and in-text citations have been updated to ensure consistency with PLOS ONE guidelines.

6. Data availability statement

The Data Availability statement has been revised to clarify that the anonymised dataset will be made publicly available in Figshare without restriction upon publication. Data are provided as Supporting Information for peer review.

7. Reviewer-suggested citations

We have reviewed all suggested citations and included relevant references where appropriate. No irrelevant or non-essential citations were added.

8. Reference list accuracy

The reference list has been carefully reviewed and updated to ensure accuracy, completeness, and consistency with in-text citations. No retracted articles are cited.

Reviewer #1

Comment 1: Update AMR mortality data

We have updated the Background section to reflect the most comprehensive available estimates, including both direct (1.27 million) and associated (4.95 million) AMR-related deaths globally in 2019, based on the latest global analysis.

Comment 2: Expand conclusions beyond governance gaps

The Conclusion has been revised to incorporate additional key findings, including:

• Substantial use of broad-spectrum antibiotics across multiple indications

• Financial barriers to completing antibiotic courses

• The central role of pharmacists in antibiotic selection

This provides a more comprehensive summary of the study’s implications.

Reviewer #2

Comment 1: General feedback

We thank the reviewer for their positive assessment of the manuscript.

Comment 2: Study design clarification

The study design has been revised from “prospective descriptive cross-sectional” to “cross-sectional study”, consistent with STROBE guidelines and the analytical nature of the study.

Comment 3: Sampling justification

We have clarified that all eligible dispensing events during the study period were included consecutively to minimise selection bias. We have also stated that a formal sample size calculation was not performed due to the observational design and defined study period.

Comment 4: Clarification of “both” dispensing source

We have clarified that cases classified as “both” refer to instances where a prescription was presented but additional antibiotics were requested and supplied without prescription.

Comment 5: Multiple antibiotics per prescription

We have added detail to the Results section specifying that 52 (10.3%) dispensing events involved more than one antibiotic, most commonly involving two antibiotics, with very few involving three or more.

Comment 6: Inclusion of AWaRe classification in results

We have incorporated the WHO AWaRe classification into the Results section, noting that the majority of antibiotics dispensed belonged to the Access and Watch categories, with frequent use of Watch antibiotics such as macrolides and fluoroquinolones. This has also been further discussed in the Discussion section.

Additional Revisions

In addition to the points raised by reviewers and editors, we have undertaken further refinements to improve clarity and internal consistency:

• Table 2 has been fully revised to ensure accurate, dataset-derived classification of antibiotics by clinical indication using consistent antibiotic class groupings.

• The Results and Discussion sections have been updated to ensure complete alignment with the revised table and underlying dataset.

• Minor language edits have been made throughout to improve clarity, precision, and readability.

We believe these revisions have significantly strengthened the manuscript and addressed all concerns raised during the review process. We are grateful for the opportunity to revise our work and hope it is now suitable for publication in PLOS ONE.

Yours sincerely,

Eleanor Turnbull-Jones

(on behalf of all authors)

---

## [Decision Letter · Decision Letter 1]

30 Apr 2026

Antibiotic dispensing practices and antimicrobial stewardship gaps in community pharmacies in Kakamega County, Kenya

PONE-D-26-02714R1

Dear  Ms Eleanor Rachel Turnbull-Jones,

We’re pleased to inform you that your manuscript has been judged scientifically suitable for publication and will be formally accepted for publication once it meets all outstanding technical requirements.

Kind regards,

Sarah Nanzigu, Ph.D.,MSc.,MBchB

Academic Editor

PLOS One

Additional Editor Comments (optional):

Reviewers' comments:

Reviewer's Responses to Questions

**Comments to the Author**

1. If the authors have adequately addressed your comments raised in a previous round of review and you feel that this manuscript is now acceptable for publication, you may indicate that here to bypass the “Comments to the Author” section, enter your conflict of interest statement in the “Confidential to Editor” section, and submit your "Accept" recommendation.

Reviewer #1: (No Response)

Reviewer #2: All comments have been addressed

2. Is the manuscript technically sound, and do the data support the conclusions?

Reviewer #1: Yes

Reviewer #2: Yes

3. Has the statistical analysis been performed appropriately and rigorously? 

Reviewer #1: Yes

Reviewer #2: Yes

4. Have the authors made all data underlying the findings in their manuscript fully available?

Reviewer #1: Yes

Reviewer #2: Yes

5. Is the manuscript presented in an intelligible fashion and written in standard English?

Reviewer #1: Yes

Reviewer #2: Yes

6. Review Comments to the Author

Reviewer #1: check all grammatical errors and editing mistakes in final manuscript, improve the quality of fig.1, recheck the citations once again

Reviewer #2: review comments addressed. They did not include AWARE category in modifying result. But, they have mentioned about that briefly in discussion

7. PLOS authors have the option to publish the peer review history of their article (what does this mean?). If published, this will include your full peer review and any attached files.

Reviewer #1: **Yes:** Dr. Saurabh Singh

Reviewer #2: No

---

## [Editor Report · Acceptance letter]

PONE-D-26-02714R1

PLOS One

Dear Dr. Turnbull-Jones,

I'm pleased to inform you that your manuscript has been deemed suitable for publication in PLOS One. Congratulations! Your manuscript is now being handed over to our production team.

Kind regards,

on behalf of

Dr. Sarah Nanzigu

Academic Editor

PLOS One